# Model-based inference of neutralizing antibody avidities against influenza virus

**Janina Linnik**[1,2], **Mohammedyaseen Syedbasha**[3], **Yvonne Hollenstein**[3], **Jörg Halter**[4], **Adrian Egli**[3,5]*, **Jörg Stelling**[1,2]*

**1** Department of Biosystems Science and Engineering, ETH Zurich, Basel, Switzerland, **2** Swiss Institute for Bioinformatics, Basel, Switzerland, **3** Department of Biomedicine, University of Basel, Basel, Switzerland, **4** Division of Hematology, University Hospital Basel, Basel, Switzerland, **5** Division of Clinical Bacteriology and Mycology, University Hospital Basel, Basel, Switzerland

* adrian.egli@usb.ch (AE); joerg.stelling@bsse.ethz.ch (JS)

**Data Availability Statement:** The authors confirm that all data underlying the findings are fully available without restriction. The model is available in the R package himodel (https://gitlab.com/csb.ethz/himodel). R code including figure-generating

## Abstract

To assess the response to vaccination, quantity (concentration) and quality (avidity) of neutralizing antibodies are the most important parameters. Specifically, an increase in avidity indicates germinal center formation, which is required for establishing long-term protection. For influenza, the classical hemagglutination inhibition (HI) assay, however, quantifies a combination of both, and to separately determine avidity requires high experimental effort. We developed from first principles a biophysical model of hemagglutination inhibition to infer IgG antibody avidities from measured HI titers and IgG concentrations. The model accurately describes the relationship between neutralizing antibody concentration/avidity and HI titer, and explains quantitative aspects of the HI assay, such as robustness to pipetting errors and detection limit. We applied our model to infer avidities against the pandemic 2009 H1N1 influenza virus in vaccinated patients (n = 45) after hematopoietic stem cell transplantation (HSCT) and validated our results with independent avidity measurements using an enzyme-linked immunosorbent assay with urea elution. Avidities inferred by the model correlated with experimentally determined avidities ($\rho = 0.54$, 95% CI = [0.31, 0.70], $P < 10^{-4}$). The model predicted that increases in IgG concentration mainly contribute to the observed HI titer increases in HSCT patients and that immunosuppressive treatment is associated with lower baseline avidities. Since our approach requires only easy-to-establish measurements as input, we anticipate that it will help to disentangle causes for poor vaccination outcomes also in larger patient populations. This study demonstrates that biophysical modelling can provide quantitative insights into agglutination assays and complement experimental measurements to refine antibody response analyses.

## Author summary

Influenza vaccines are assessed based on the induced antibody response, where antibody quantity (concentration) and antibody binding strength (avidity) determine the potency to neutralize the virus. In addition, an increase in avidity indicates a successful germinal center reaction, which is required for establishing long-term protection. However, the

scripts are available on GitLab (https://gitlab.com/csb.ethz/himodel-manuscript).

**Funding:** We acknowledge funding by the Swiss National Science Foundation (SystemsX iPhD 2014/246 to AE and JS and SNSF Ambizione Score PZ00P3_154709 to AE). The funders had no role in study design, data collection and analysis, decision to publish, or preparation of the manuscript.

**Competing interests:** The authors have declared that no competing interests exist.

hemagglutination inhibition (HI) assay—traditionally used to assess influenza vaccines—measures a combination of both antibody concentration and avidity, and to separately determine avidity requires high experimental effort. We developed a biophysical model of the HI assay, which enables the inference of antibody avidities from measured HI titers and antibody concentrations. We applied our approach to a vaccinated population of immunocompromised patients after blood stem cell transplantation and validated our results experimentally. The model predicted that vaccination induced an increase in avidity in only a few patients and that patients under immunosuppressive treatment show lower baseline avidities. Since our approach requires only easily measurable data as input, it can facilitate the investigation of vaccine responses in larger populations. This study demonstrates that biophysical modelling can complement experimental data and provide additional details on agglutination experiments and antibody responses.

## Introduction

To assess influenza vaccine efficacy, hemagglutination inhibition (HI) titers are traditionally used as a surrogate for the influenza-neutralization capacity of vaccine-induced antibodies in serum [1, 2]. The HI assay makes use of the phenomenon that influenza viruses bind with their surface receptor hemagglutinin (HA) to red blood cells (RBCs) and can cross-link them to macroscopic cell aggregates in a process called hemagglutination [3]. In the presence of influenza-binding antibodies that block RBC binding sites, hemagglutination is inhibited. This allows quantifying the neutralization capacity of serum antibodies in dilution experiments: the highest serum dilution that fully inhibits hemagglutination is determined, and its dilution factor is reported as the HI titer [4].

The HI titer measures a combination of both antibody concentration and avidity, where avidity quantifies the overall strength of a multivalent antibody binding to hemagglutinin epitopes involved in virus-RBC interaction (neutralizing binding). When assessing vaccine response, however, it is important to distinguish between antibody concentration and avidity. In particular, no increase in avidity following vaccination indicates a hampered formation of germinal centers (GCs) where B cells undergo affinity maturation and proliferate to long-lived B cells, providing long-term protection [5, 6].

Avidity measurements of serum antibodies are time-consuming and costly. Commonly used techniques are surface plasmon resonance (SPR) and elution experiments with chaotropic agents (such as urea) based on enzyme-linked immunosorbent assays (ELISAs). While SPR experiments require special equipment and long calibration, elution assays are very sensitive to experimental conditions, and optimal protocols might vary substantially for different samples [7, 8]. In comparison, measurements of HI titers and serum IgG concentrations are faster to establish and simpler to perform [9]. HI assays are considered the gold standard and routinely performed in vaccine studies; they proved to be fast, cheap, and reliable. IgG concentrations can be determined in standardized ELISA experiments. These are suitable for large-scale serological studies because they can be fully automated and yield highly reproducible results. Therefore, estimation of avidities from HI titers and IgG concentrations would facilitate influenza antibody response analyses in larger populations.

Here, we present a biophysical model of the HI assay that mechanistically describes the relationship of neutralizing IgG concentration and avidity to the resulting HI titer, and enables the inference of neutralizing IgG avidities from HI titers and ELISA-detected IgG concentrations. We applied our approach to vaccinated hematopoietic stem cell transplantation (HSCT)

patients, focusing on IgG antibodies specific to pandemic influenza A/California/7/2009 (H1N1pmd09). Despite available vaccines, the case fatality rate for influenza is 17–29% in these patients [10]. HSCT patients are commonly immunocompromised due to post-transplant immune reconstitution and immunosuppressive treatment against graft versus host disease. Since patients with low antibody avidities are at risk for fatal infections, we investigated the association of inferred avidities with three indicators of immunocompromised status as defined by CDC [11], known to be associated with immune cell proliferation, affinity maturation and antibody production [12]: first two years post transplantation, immunosuppressive treatment, and chronic graft-versus-host disease (cGVHD) grade according to NIH criteria [13]. Our model detected that immunosuppressive treatment is associated with lower baseline avidities, but we did not detect a significant association with cGVHD or the time after transplantation. In addition, our model suggests that vaccination induced affinity maturation of neutralizing antibodies in only a few patients.

## Results

### Model of the hemagglutination inhibition (HI) assay

We extended models of antibody-virus interaction [14] and cell-cell agglutination [15] to a model that mechanistically captures the key processes of the HI assay (Fig 1). The HI assay is performed in three consecutive steps [4]: (i) Serial dilution of patient serum and 30 min incubation with influenza virus, (ii) addition of RBCs followed by 30 min incubation, and (iii) determination of the HI titer based on the presence or absence of hemagglutination inhibition in each serum dilution (Fig 1, top). We represent these steps separately: the model output of one step serves as input for subsequent steps (Fig 1, bottom).

**Step 1 (binding of antibodies to virus).**   We modeled the binding of IgG antibodies to virus hemagglutinin (HA) as a diffusion-controlled reversible reaction between IgG molecules and virus particles (see Methods for details). Each homotrimeric HA receptor has three identical binding sites for monoclonal IgG, but we assume that one HA trimer accommodates at most one IgG molecule due to steric hindrance [16–18]. Serum contains a mixture of polyclonal IgG antibodies. Thus, after the addition of influenza virus to serum, HA-specific IgG clones form a mixture of IgG-HA complexes according to their individual dissociation constants (avidities). We assume that any other interactions are negligible because serum samples are pretreated with receptor destroying enzyme (RDE) to limit unspecific binding. We consider the total concentration of HA-specific IgG and the apparent dissociation constant $K_D^{app}$, which is proportional to the ratio of free HA-specific IgG molecules over all formed IgG-HA complexes at equilibrium. Its inverse $1/K_D^{app}$ is interpreted as the apparent serum IgG avidity. We compute the fraction of antibody-bound virus at binding equilibrium for each serum dilution (Fig 1, left) as input for step 2 because 30 min incubation suffices to reach binding equilibrium.

**Step 2 (hemagglutination).**   When RBCs are added, virus particles bind reversibly with free HA binding sites to sialic acid (SA) linked receptors on RBCs. We assume that IgG antibodies and SA-linked receptors do not compete for HA binding sites because the affinity of SA to HA is in the mM range [19–21], far below the affinity of HA-specific IgGs in the nM range [22]. The tight binding of the virus to RBCs results from binding multiple SA moieties simultaneously [23]. The virus-RBC interactions will eventually induce hemagglutination. We model it as a coagulation process [24], where RBCs stick together whenever they collide such that virus particles can cross-link them. Only when a free SA-linked receptor on an RBC meets a free HA on a virus particle that is simultaneously bound to another RBC, the collision leads to a successful cross-link. We define a degree of hemagglutination that takes the value 0% without

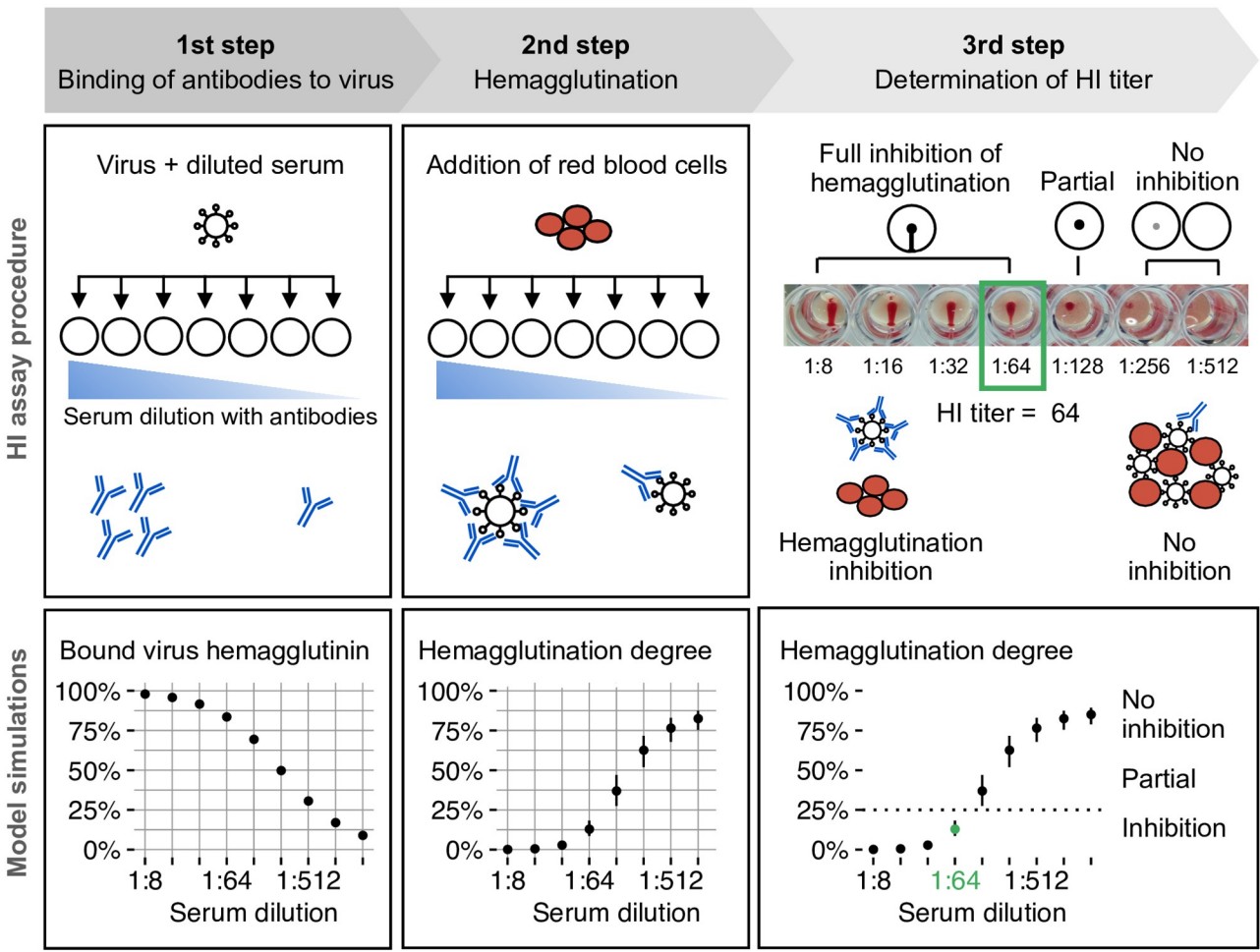

**Fig 1. Overview of the hemagglutination inhibition (HI) assay (top) and illustrative simulation results (bottom).** First step: patient serum is serially diluted and incubated with a constant amount of influenza virus. The model computes the amount of antibody-bound viral hemagglutinin (HA) for each serum dilution. Second step: red blood cells (RBCs) are added to each dilution and virus particles with free HA binding sites cross-link RBCs to cell aggregates. The model predicts a switch-like increase in agglutinated RBCs with decreasing antibody concentration. Third step: the plate is tilted by 90 degrees to detect full hemagglutination inhibition. If none/few RBCs are agglutinated, sedimented RBCs flow down to the rim. By definition, those wells show full hemagglutination inhibition. The reciprocal of the maximal inhibitory dilution is the HI titer. We classify our simulation results into inhibition and no inhibition by setting a threshold at 25% hemagglutination. Simulation results show median and interquartile range indicating the uncertainty due to experimental conditions (RBC concentration, virus concentration, readout time) and model parameters (summarized in Table 1) for an IgG serum concentration of 25 nM (4 μg/mL) and $K_D^{app} = 0.1$ nM.

any hemagglutination (not a single cross-link), and 100% when all RBCs form a single aggregate. The model predicts a switch-like increase in the degree of hemagglutination with decreasing antibody concentration, consistent with the experimentally observed switch-like behavior of the assay (Fig 1, middle).

**Step 3 (determination of HI titer).** After another 30 min incubation, each serum dilution is inspected for hemagglutination inhibition, and the reciprocal of the maximal dilution that shows full inhibition is the HI titer (Fig 1, right). To model this binary decision (inhibition or no inhibition), we classify the outcome by setting a threshold at 25% hemagglutination because we define 50% hemagglutination as partial inhibition and our model predicts for $\geq 1$ HA unit virus a hemagglutination degree of $\geq 75\%$. By definition, this is interpreted as full inhibition

**Table 1. Model parameters and variables.** The assumed ranges of uncertainty and biological variability in model parameters and variables are defined by the distributions used in the sensitivity analysis. Abbreviations are: IgG, Immunoglobulin G; RBC, red blood cell; HA, hemagglutinin; HAU, HA unit; SA, sialic acid.

| Description | Symbol | Value | Distribution in sensitivity analysis | Reference |
|---|---|---|---|---|
| Serum IgG concentration | $A_0$ | Sample-specific | Unif(0, 2800) nM (0–420 μg/mL) | [25] |
| Apparent IgG dissociation constant | $K_D^{app}$ | Sample-specific | Unif(0.001, 300) nM | [22] |
| Initial virus concentration | $V_0$ | $1.3 \cdot 10^{-4}$ nM (4 HAU) | Unif($0.9 \cdot 10^{-4}$, $2.3 \cdot 10^{-4}$) nM (3–7 HAU) | [4] |
| Initial RBC concentration | $RBC_0$ | $3.1 \cdot 10^{-5}$ nM | Unif($1.6 \cdot 10^{-5}$, $6.3 \cdot 10^{-5}$) nM | [4] |
| Number of HA receptors per virus | $r$ | 400 | Discrete Unif(300,500) | [26, 27] |
| Number of epitopes per HA receptor | $e$ | 3 | Fixed at 3 | [28, 29] |
| Number of shaded epitopes per bound IgG | $e^*$ | 3 | Bernoulli(0.5) with $e^* \in \{3, 6\}$ | [18] |
| Number of SA receptors per RBC | $b$ | $4.5 \cdot 10^5$ | Discrete Unif($4 \cdot 10^5$, $5 \cdot 10^5$) | [30, 31] |
| Number of shaded SA receptors per bound virus | $b^*$ | 34 | Sampled from model | See Methods |
| SA-HA association rate constant | $k_{ass}^{RBC}$ | $2 \cdot 10^{-6}\ s^{-1}$ | Lognorm(log($2 \cdot 10^{-6}$), $0.2^2$) $s^{-1}$ | [23] |
| SA-HA dissociation rate constant | $k_{diss}^{RBC}$ | $2 \cdot 10^{-4}\ nM^{-1}s^{-1}$ | Lognorm(log($2 \cdot 10^{-4}$), $0.2^2$) $nM^{-1}s^{-1}$ | [23] |
| RBC agglutination rate constant | $k_{agg}$ | $2 \cdot 10^6\ s^{-1}$ | Unif($0.4 \cdot 10^6$, $13 \cdot 10^6$) $s^{-1}$ | Estimated from data |

(S1C Fig), suggesting that differences in hemagglutination degree below 25% or above 75% cannot be distinguished by eye.

We extracted parameters and their uncertainty ranges from literature for IgG, chicken RBCs and influenza virus (Table 1). In addition, we established the agglutination rate parameter from hemagglutination inhibition experiments with a serum sample from a healthy volunteer (see Methods). Next, we investigated the impact of the assumed parameter ranges on the simulated hemagglutination degree in a global sensitivity analysis to investigate the robustness of model predictions to model assumptions and experimental variability.

## Sensitivity analysis shows model's robustness to uncertainties in parameters and experimental conditions

To infer antibody avidities accurately, the model needs to be sensitive to the experimental data used as inputs. However, it should not be sensitive to other experimental factors and uncertainties in model parameters. To evaluate the model in this respect, we used Sobol sensitivity analysis [32], which attributes variance in model output (here: hemagglutination degree) to the individual model input factors. The more influential the input factor is, the higher is its contribution to the variance in hemagglutination degree. We considered the ranges for all model input factors summarized in Table 1. Specifically, for IgG concentration and avidity the ranges match the experimentally observed ranges for H1N1pmd09-specific IgG after vaccination in adults [22, 25]. For experimental conditions, we aimed to generously cover experimental variability. For model parameters, we considered measurement uncertainty and biological variability as described in the literature. The kinetic rate constants describing the association and dissociation of HA and sialic acid were not available for H1N1pmd09. These parameters were sampled from log-normal distributions (Table 1) in agreement with reported values for HA from another influenza strain [23]. These values correspond to a log-normally distributed dissociation constant centered at 100 nM and covering a range of approximately 30 nM to 300 nM. This broad range also accounts for widely varying binding constants across influenza strains [33].

Sensitivity analysis showed that serum IgG avidity and concentration are the most influential factors for hemagglutination (Fig 2A). Variability in RBC and virus concentration and in readout time (30–45 min) contribute very little to the total variance. The model is also robust to uncertainty in all model parameters except for the kinetic agglutination rate of RBCs ($k_{agg}$),

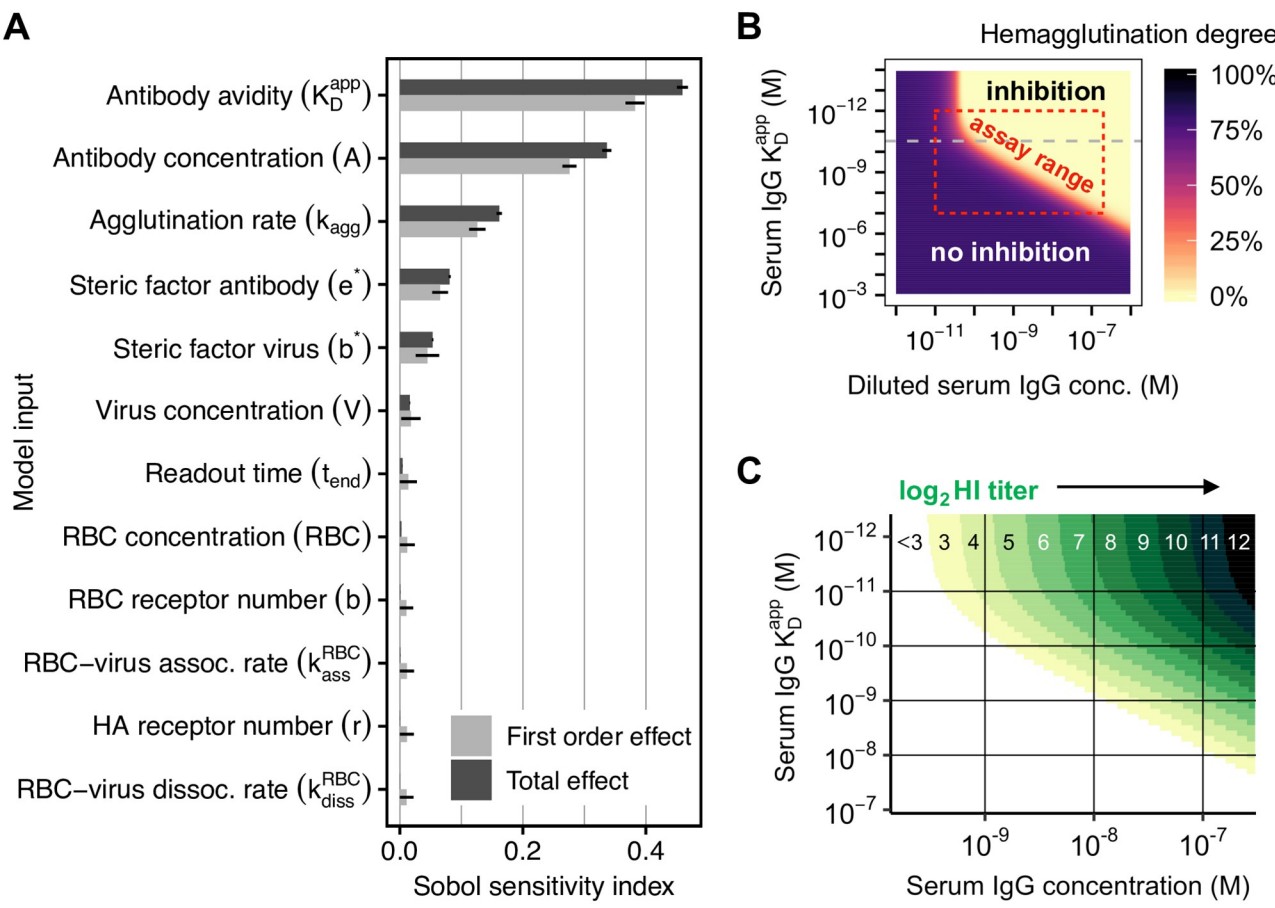

**Fig 2. Model sensitivity and resolution of the hemagglutination inhibition assay for influenza H1N1pdm09.** (**A**) Sensitivity analysis using Sobol indices. First-order effects show only the linear contribution to the total variance in hemagglutination degree (they sum up to 1), whereas total effects consider also interactions (see Methods for details). (**B**) Predicted degree of hemagglutination for different IgG concentrations and apparent dissociation constants $K_D^{app}$. The red box indicates the usual assay range, bounded by the biological range of $K_D^{app} = 0.001 - 100$ nM, and the gray dashed line indicates $K_D^{app} = 0.03$ nM. (**C**) Predicted HI titers for the biological range of influenza-specific serum IgG and $K_D^{app}$. Colored areas correspond to titers shown on top.

which we varied within the 95% highest probability density interval estimated from our calibration data. Other relevant factors were the ability of IgG to bind two HA receptors simultaneously and the number of RBC receptors that are covered by one bound virion. Hence, the model's predictions are dominated by the measured input quantities, despite uncertainties in experimental conditions, mechanisms, and parameters.

The model predicts a clear separation between hemagglutination inhibition and no inhibition—partial inhibition occurs only within a small range of IgG concentration and avidity (Fig 2B). Thus, the well-known binary nature of the assay is captured. Using an initial virus concentration of 4 HA units as defined by WHO ensures both high sensitivity and robustness, whereas 8 HA units or more increase robustness but lower sensitivity (S1E Fig). The model predicts also a yet unknown property of the HI assay: for avidities $K_D^{app} \geq 0.03$ nM, hemagglutination quantifies a combination of IgG concentration and avidity, but for very high avidities $K_D^{app} < 0.03$ nM, the assay only detects changes in IgG concentration (Fig 2B).

Within the linear range for $K_D^{app} \geq 0.03$ nM, a doubling in IgG concentration or avidity results in a doubling of the predicted HI titer (Fig 2C). In other words, a two-times lower antibody avidity can be compensated by a two-times higher antibody concentration. However,

this only applies to the linear range and the exact relationship depends on the considered avidity and concentration ranges (Fig 2C). The model also suggests why HI titers above 8192 (= 13 in $\log_2$) are rarely observed. Even for a high serum IgG concentration of 1000 nM (150 μg/mL), such high titers require antibody avidities in the fM range, but influenza-specific antibody affinities in vaccinated healthy adults lie in the nM range [22].

In summary, we conclude that the model yields robust simulation results in the applicable assay range, and reveals new quantitative aspects of the HI assay: The predicted hemagglutination degree is not sensitive to uncertainty in model parameters and variability in experimental conditions. It is mostly determined by the avidity and concentration of serum antibodies (Fig 2A). In conclusion, the model enables to quantitatively relate IgG concentration and avidity to HI titer.

## Inference of neutralizing antibody avidities in HSCT patients

Next, we applied our model to infer avidities from ELISA-detected serum IgG concentrations and HI titers specific to H1N1pdm09 in HSCT patients (patient characteristics are summarized in Table 2). We used a Bayesian approach that accounts for uncertainties due to ELISA measurement error and discretization in HI titers (see Methods for details). Model parameters related to RBCs and virus particles were fixed for all serum samples (Table 1), because our sensitivity analysis (Fig 2A) suggested that differences in HI titer arise mostly from differences in serum-specific IgG concentration and avidity. All patients received two doses of non-adjuvanted trivalent seasonal influenza vaccine on d0 and d30 (see Methods). Measurements were available from 45 patients at five time points before (d0) and after (d7, d30, d60, d180) the first vaccination with 221 serum samples in total. HI titers and IgG concentrations were significantly correlated (Kendall's $\tau = 0.69$, $P < 10^{-15}$, rank correlation for ordinal data; Fig 3A).

For serum samples with HI titers below assay resolution (HI titer $< 8$), we could only infer an upper bound for the avidity (it could be lower, but not higher). This affected 23 serum samples from seven patients. Analogously, for serum samples with $K_D^{app} \leq 0.03$ nM, we could, in principle, only report a lower bound, but all inferred avidities for our patient cohort exceeded this threshold. In 24 samples, inferred $K_D^{app}$-values showed very large uncertainty (approximately ±100%) due to large measurement error in ELISA measurements; we excluded these samples from further analysis. In the remaining samples, posterior distributions of inferred avidities were log-normally distributed (S2 Fig) and we determined the uncertainty intervals due to discretized HI titer measurements and ELISA measurement error by sampling, yielding an average uncertainty in $K_D^{app}$-values of approximately ±30% (range 20–57%, interquartile range (IQR) 25–30%).

In summary, we were able to infer 197 avidities from in total 43 patients (89% of analyzed samples). The inferred avidities ranged from $K_D^{app} = 0.1$ nM to $\geq 22$ nM (upper bound), with a median of 1.7 nM and IQR 0.9–2.5 nM. Inferred avidities and HI titers were significantly correlated (Kendall's $\tau = 0.56$, $P < 10^{-15}$), although the correlation was weaker than for IgG concentration (Fig 3A).

## Inferred avidities correlate with experimentally determined avidities

We validated our model with experimental avidity measurements of 59 serum samples from 12 patients. We performed ELISA-based elution assays that quantify the fraction of IgG remaining bound after 3h incubation with 4M urea, yielding a measure for the overall binding strength of serum IgG to H1N1pmd09 in the form of an avidity index between 0 (low avidity) and 1 (high avidity).

**Table 2. Characteristics of allogeneic hematopoietic stem cell transplant patients (all patients and subset of patients with experimentally determined avidities for validation of inferred avidities).** Abbreviations are: IQR, interquartile range; GVHD, graft-versus-host disease.

| | | All | Experimentally validated subset |
|---|---|---|---|
| Total | | 45 | 12 |
| Age | Median, IQR (years) | 58, 44–64 | 45, 43–67 |
| | ≥ 65 years | 11 (25%) | 4 (33%) |
| Sex | Male | 23 (51%) | 4 (33%) |
| | Female | 22 (49%) | 8 (67%) |
| Underlying disease | Acute myeloid leukemia (AML) | 17 (38%) | 6 (50%) |
| | Acute lymphoblastic leukemia (ALL) | 8 (18%) | 2 (17%) |
| | Chronic myeloid leukemia (CML) | 5 (11%) | 1 (8%) |
| | Chronic lymphocytic leukemia (CLL) | 5 (11%) | 1 (8%) |
| | Multiple myeloma (MM) | 5 (11%) | 1 (8%) |
| | Plasma cell leukemia (PCL) | 1 (2%) | 0 |
| | Myeloproliferative neoplasms (MPN) | 1 (2%) | 1 (8%) |
| | Myelodysplastic syndromes (MDS) | 2 (4%) | 0 |
| | Non-Hodgkin lymphoma (NHL) | 1 (2%) | 0 |
| Time after transplantation | median, IQR (years) | 4, 2–8 | 6, 3–8 |
| | 1–2 years | 16 (36%) | 2 (17%) |
| | 3–5 years | 13 (29%) | 3 (25%) |
| | > 5 years | 16 (36%) | 7 (58%) |
| Transplant source | Peripheral blood | 40 (89%) | 11 (92%) |
| | Bone marrow | 5 (11%) | 1 (8%) |
| Donor source | Matched related donor | 16 (36%) | 5 (42%) |
| | Matched unrelated donor | 21 (47%) | 5 (42%) |
| Disease status[a] | Complete remission | 42 (93%) | 12 (100%) |
| | Stable | 1 (2%) | 0 |
| | Recurrence | 2 (4%) | 0 |
| | Progressive | 0 | 0 |
| Immunosuppression[a] | None | 18 (40%) | 2 (17%) |
| | Mild (grade 1) | 6 (13%) | 2 (17%) |
| | Moderate (grade 2) | 14 (31%) | 6 (50%) |
| | Severe (grade 3) | 7 (16%) | 2 (17%) |
| Immunosuppressive treatment[a] | Prednisone | 13 (29%) | 5 (42%) |
| | Tacrolimus | 14 (31%) | 7 (58%) |
| | Mycophenolate | 9 (20%) | 3 (25%) |
| | Cyclosporine A | 4 (9%) | 0 |
| | Rituximab[b] | 3 (7%) | 0 |
| Chronic GVHD | None | 15 (33%) | 0 |
| | Mild (grade 1) | 9 (20%) | 4 (33%) |
| | Moderate (grade 2) | 10 (22%) | 6 (50%) |
| | Severe (grade 3) | 11 (24%) | 2 (17%) |

[a]Before vaccination,

[b]within the previous six months.

The inferred and the experimentally determined avidities were significantly correlated (Pearson's $\rho$ = 0.54, 95% CI = [0.31, 0.70], $P < 10^{-4}$, Fig 3B). We detected one outlier patient (standardized residuals ≈ 3) whose serum did not show HI activity at any time point (Fig 3B), suggesting that the ELISA detected non-neutralizing IgG in this patient. Experimental and

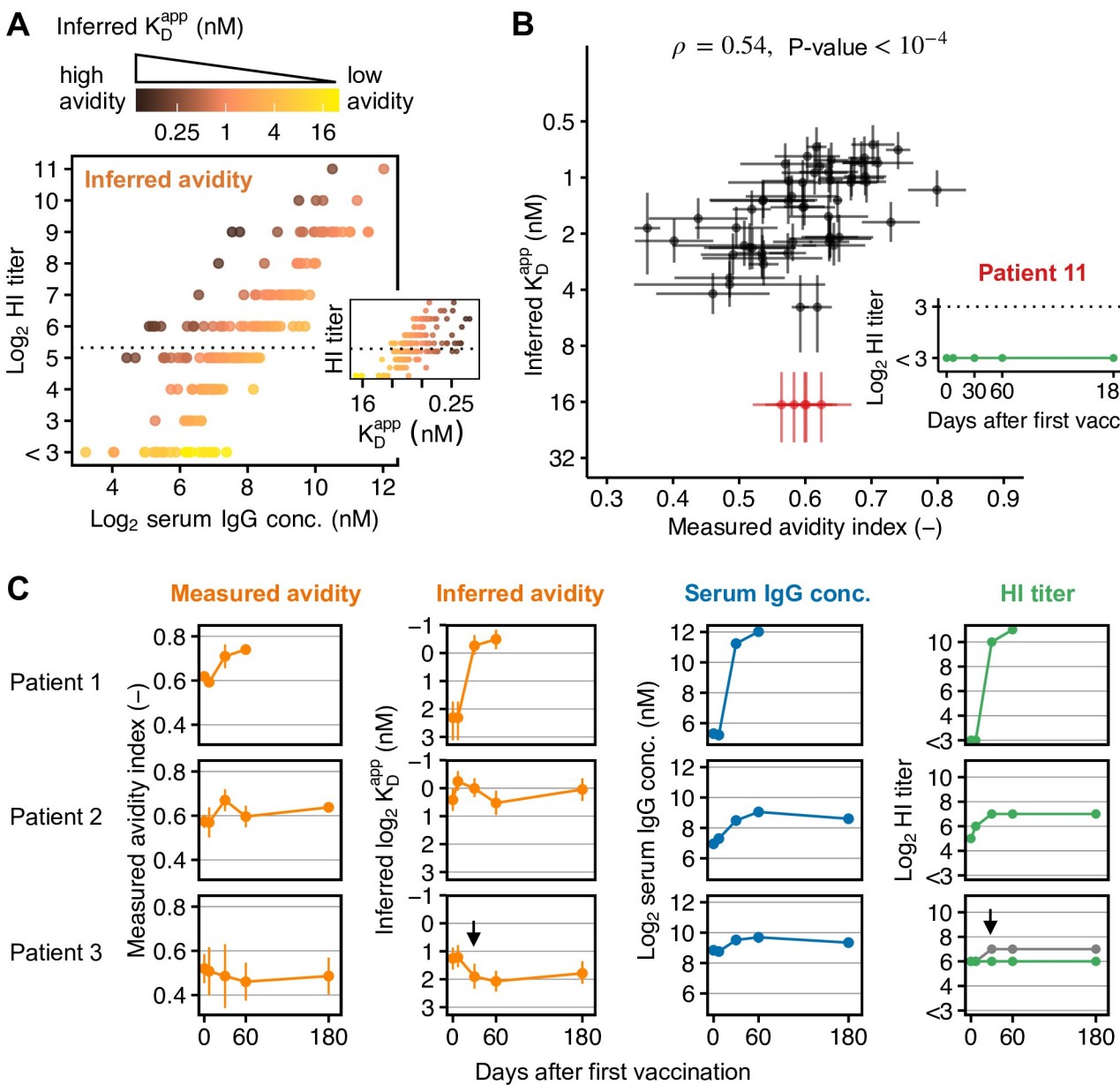

**Fig 3. Inference of antibody avidities in HSCT patients.** (**A**) ELISA-detected anti-H1N1pmd09 serum IgG concentration, HI titers and corresponding inferred apparent dissociation constants $K_D^{app}$ from 197 serum samples from 43 HSCT patients. The dashed line indicates the seroprotection threshold (HI titer $\geq$ 40). (**B**) Correlation of inferred and experimentally determined avidities in 59 serum samples from 12 HSCT patients (Pearson's $\rho = 0.54$, 95% CI = [0.31, 0.70]). Data show mean and standard deviation for avidity indices from two experiments (each performed in duplicates) and the median of the posterior distribution with the uncertainty range due to discretized HI titer measurements and ELISA measurement error for inferred $K_D^{app}$-values (see Methods for details on inference and S2 Fig for posterior distributions). Avidity indices correspond to the fraction of H1N1pmd09-specific serum IgG remaining bound after 4M urea treatment. Patient 11 was identified as an outlier, probably because ELISA detected non-neutralizing IgG; the patient showed no HI activity at any time point. For serum samples without detected HI activity (HI titer < 8; all five samples from patient 11 and three additional samples from in total two patients), the measured avidity index is plotted against the estimated upper bound for the inferred avidity and the uncertainty interval reflects the estimated uncertainty of the upper bound due to discretized HI titer measurements and ELISA measurement error. (**C**) Example patients with different types of responses to vaccination. In patient 3, we detected an increase in non-neutralizing IgG on d30. The predicted HI titer for the observed increase in IgG (shown in gray) is twice as high as the actually observed titer (green). For all 12 patients with experimentally determined avidities see S3 Fig.

inferred avidities distinguished different types of patient responses, for example, where both IgG concentration and avidity increased after vaccination (patient 1 in Fig 3C) or where an increase in HI titer was mostly explained by an increase in IgG concentration (patient 2). We identified one patient that produced non-neutralizing IgG (patient 3): Here, the ELISA detected an increase in IgG concentration that leads to HI titer doubling according to our model predictions. However, the HI titer did not increase at any time point (Fig 3C), suggesting that the ELISA-detected IgGs had no HI activity. The inferred $K_D^{app}$-value refers to neutralizing IgG-virus interactions only, and its value is biased towards lower avidity if the measured IgG concentration also includes non-neutralizing IgGs.

Thus, apparent serum avidities inferred by our model-based approach were in good accordance with experimentally determined avidities. However, if non-neutralizing IgG dominates in serum, the results are not directly comparable because the inferred avidity refers to neutralizing IgG with HI activity.

## Detection of vaccine-induced affinity maturation in HSCT patients

Next, we compared the vaccine-induced increase in inferred avidities in all investigated HSCT patients and identified candidates for successful GC formation and affinity maturation (Fig 4). Since the establishment of GCs takes approximately seven days [6], we considered an increase in IgG concentration and avidity on d30 or d60 as indicative for GC formation (patients were vaccinated on d0 and d30; see Methods).

Given the uncertainty in inferred $K_D^{app}$-values, we could detect fold changes in avidity of approximately $> 1.5$ or $< 0.5$ (except for samples below assay resolution with HI titer $< 8$) (S4A Fig). Eight patients showed a detectable increase in avidity on d30 and/or d60, of which only one showed no increase in serum IgG (Fig 4B). This suggests that vaccination induced GC formation and affinity maturation in seven patients (including patient 1 in Fig 3C). Serum avidity returned back to baseline on d180 in most of these patients, suggesting that vaccination failed to induce a sustained production of high-avidity antibodies. Over all patients showing a detectable increase in avidity at any time point after vaccination (n = 11), we observed a time-dependent increase with the largest increase on d60, i.e., after the booster dose (Fig 4C), consistent with our understanding of GC dynamics [6].

In summary, although 30 patients showed an increase in serum IgG concentration on d30 and/or d60, only 7/30 patients (23%) are candidates for vaccine-induced affinity maturation, and 6/30 patients (20%) showed vaccine-induced production of non-neutralizing IgG (such as patient 3 in Fig 3C). We excluded 4/45 patients as they showed too large measurement uncertainty in ELISA-detected IgG concentration on several time points (see above).

## Association with criteria for compromised immune response

Finally, we investigated associations between inferred avidities, IgG concentration and HI titers with time post HSCT $\leq 2$ years, intake of immunosuppressive drugs quantified by immunosuppression grade ranging from 0 (none) to 3 (severe), and cGVHD grade with the same range (Fig 4D). We investigated effects on baseline (levels before vaccination) and response (relative increase) in a multivariable regression analysis with patient-specific random intercepts, controlling for sex and age. Regression was performed on log2-transformed values using a model for continuous data for avidity/concentration and a model for sequential ordinal data for HI titers [34]. When analyzing the vaccine-induced increase in avidity, we excluded non-neutralizing IgG responders (n = 6) because their inferred avidities are not indicative of affinity maturation (see Methods for details).

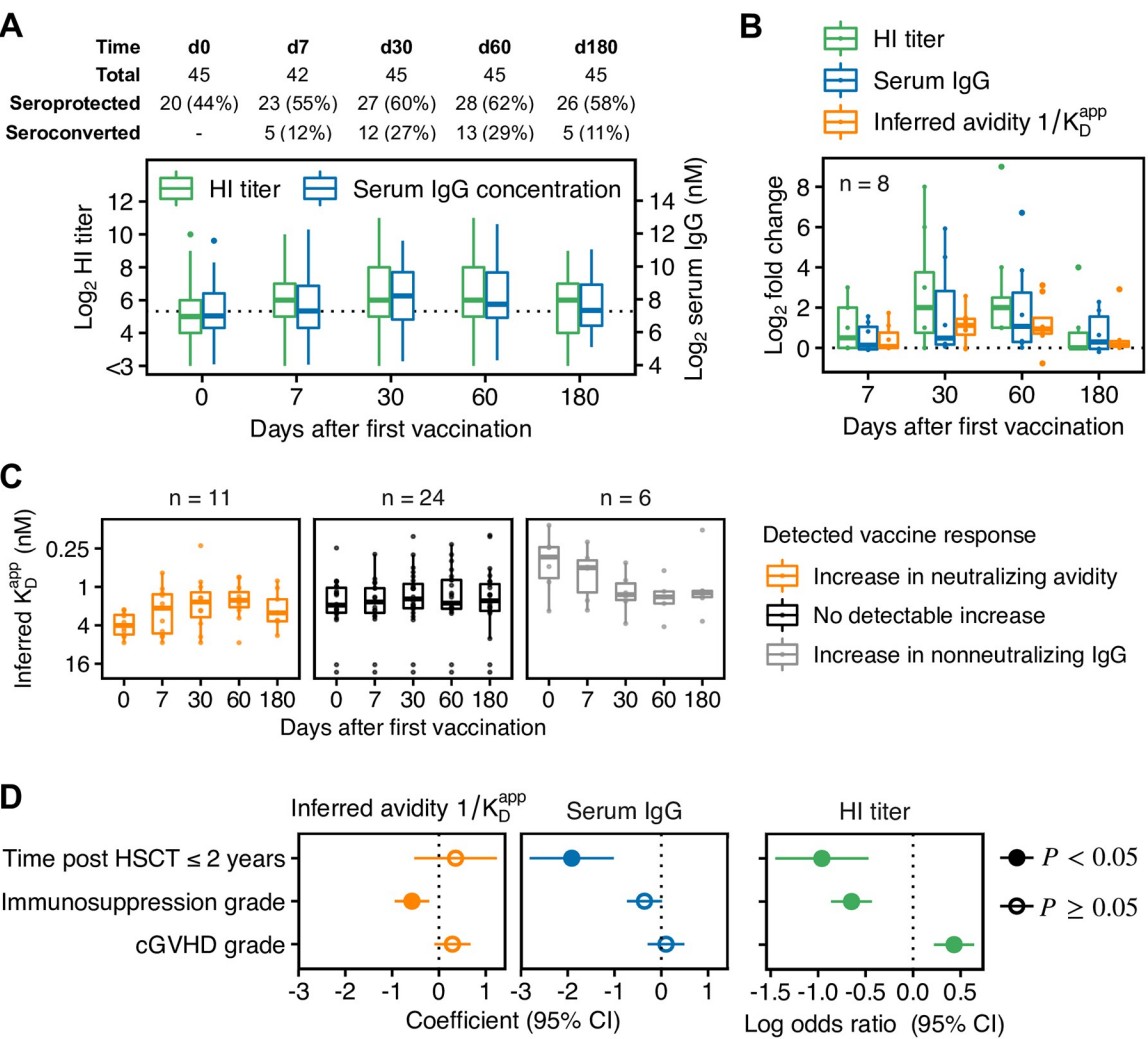

**Fig 4. Response to influenza vaccination against H1N1pmd09 in HSCT patients.** (**A**) HI titers and ELISA-detected serum IgG concentrations in the investigated patient population. Patients were vaccinated on d0 and d30 with a non-adjuvanted trivalent influenza vaccine. Seroprotection corresponds to HI titer $\geq 40$ and seroconversion to a four-fold HI titer increase compared to d0. (**B**) Fold changes in HI titer, serum IgG and inferred avidity $(1/K_D^{app})$ in all patients with a detectable increase in inferred avidity on d30 and/or d60. (**C**) Comparison of inferred avidities between patients with a detectable increase in avidity at any time point after vaccination (left), patients with no detectable increase (middle), and patients with a detectable increase in non-neutralizing IgG (right). We excluded 4/45 patients as they showed too large measurement uncertainty in IgG concentration on several time points. (**D**) Estimated effects on baseline levels of criteria for compromised immune response. Effects were estimated in a multivariable regression analysis on log2-transformed values controlling for sex and age. Time after HSCT was encoded as a binary variable (1 for HSCT $\leq 2$ years and 0 for HSCT $> 2$ years). Immunosuppression grade and cGVHD grade ranging from 0 (no immunosuppression/cGVHD) to 3 (severe immunosuppression/cGVHD) were encoded as ordered categorical variables with grade 0 as reference (see Methods for details).

Early transplant patients (time post HSCT $\leq 2$ years) showed significantly lower baseline levels in IgG concentration ($-1.92 \pm 0.46$, $P = 1.8 \cdot 10^{-4}$) and HI titer (log odds ratio $-0.96 \pm 0.25$, $P = 1.2 \cdot 10^{-4}$) than patients with HSCT $> 2$ years (Fig 4D). Since we corrected for immunosuppression and cGVHD grade, this effect could be explained by differences in patients' immune reconstitution and the number of previously received influenza vaccinations. At the time of this study, the annual influenza vaccine contained H1N1pmd09 already for four years. Therefore, it is likely that patients with HSCT $> 2$ years acquired durable H1N1pmd09-neutralizing antibodies in previous seasons. Yet, early transplant patients did

not show significantly different baseline avidities compared to patients with HSCT > 2 years (0.36 ± 0.45, $P$ = 0.43), potentially because previous vaccinations did not induce (detectable) affinity maturation. However, patients' immunization history was unknown and we could not further investigate this hypothesis. Patients under immunosuppression showed significantly lower baseline avidities with an estimated effect size of −0.58 ± 0.19 per immunosuppression grade ($P$ = 4.0 · 10$^{-3}$). This means, that patients with immunosuppression grade 2 showed approximately two-fold lower baseline avidities than patients without immunosuppression, while patients with immunosuppression grade 3 showed three- to four-fold lower baseline avidities (see also S5 Fig).

We did not detect significant differences in the vaccine-induced increase in concentration or avidity, potentially because the number of responders in the investigated HSCT population was too low (only 13 patients showed seroconversion, Fig 4A).

## Discussion

The HI assay is a well-established gold standard method, and yet, little is known on the relationship between HI titer, serum antibody concentration, and avidity. Mathematical models of cell agglutination by antibody cross-linking have been previously reported [15, 35] and applied to guide the design of immunoassays [36]. We presented an extension to a three-component system consisting of antibodies, viruses, and cells. Our model captures known properties of the HI assay and provides a biophysical explanation for why the HI assay has become the gold standard in serological studies. First, the assay is equally sensitive to both antibody concentration and avidity. Only for extremely high avidities ($K_D^{app}$ < 0.03 nM), it detects only changes in concentration. Second, the assay is robust to pipetting errors or other experimental variability in RBC and virus concentration.

The model allows the inference of neutralizing serum avidities from ELISA-detected IgG concentrations and HI titers, which are simpler, faster, and cheaper to measure than antibody avidities, especially in larger populations. In our experimental setup where HI titers were determined in two-fold serial dilutions, we were able to estimate avidities with a precision of approximately ±30%. A limitation of our approach is that we cannot distinguish whether HI titers below assay resolution (here: HI titer < 8) correspond to non-neutralizing IgG (which could potentially have high avidity but not to HA) or to neutralizing IgG below assay resolution (with low avidity or low concentration). In general, our modelling approach is only applicable to influenza strains and IgG antibodies with HI activity. Influenza strains that show poor agglutination properties, for instance, in hemagglutination titration assays, might not provide enough sensitivity and resolution. The detection of both neutralizing and non-neutralizing IgG (i.e., IgG without HI activity) can bias our results towards lower avidities; this possibility could be evaluated with SPR or calorimetry measurements. We also neglected IgM antibodies because IgMs show lower serum concentration than IgGs [37]. When IgM concentration is high while IgG's low, for example, on d7 after vaccination in naive subjects, modeling the contribution of IgM to the HI titer may be necessary. Finally, we applied our modelling approach to the HI assay specific for H1N1pdm09. Further studies are required to understand how it could be translated to other influenza strains or other agglutination inhibition assays in general.

Only a few patients showed a vaccine-induced increase in avidity, potentially because the number of responders was low or because the increase was below our detection limit (fold change < 1.5). However, we observed consistent effects: among 32 (13) patients with an increase in serum IgG on d30 or d60 (seroconversion on d60), we identified only seven (five) candidates for vaccine-induced affinity maturation. Thus, vaccine-induced increases in HI

titer were mostly explained by increased IgG concentration. These results might not apply to other populations, especially because hampered affinity maturation is likely in HSCT patients [12]. The precision of our inference was also sufficient to detect differences in baseline avidities between patients with and without immunosuppression. Excluding the two patients without HI activity at any time point from our analysis only slightly affects the detected association between immunosuppression grade and avidity ($-0.39 \pm 0.16$, $P = 0.02$).

It is unknown to which extent vaccination against H1N1pmd09 induces affinity maturation in HSCT patients and it is currently under debate whether poor vaccine-induced affinity maturation is responsible for the poor effectiveness of seasonal influenza vaccines [38–40]. Previous studies in healthy adults showed that serum avidity against HA1 peaks at 21–28d after vaccination and decreases almost back to baseline on d180 [25, 41]. We observed a similar behavior among those HSCT patients that showed a detectable increase in avidity, although, in contrast to healthy subjects, HSCT patients received a booster dose on d30. We observed the largest increase in avidity on d60, suggesting that the booster dose might enhance vaccine-induced affinity maturation.

Interestingly, all patients identified as non-neutralizing IgG producers showed relatively high HI titers (geometric mean titer = 128, range = 32–1024) and high neutralizing avidities before vaccination (Fig 4C). Even if we detected both neutralizing and non-neutralizing IgG on d0 in these patients, this would bias the inferred $K_D^{app}$-values towards lower avidities, meaning that the actual neutralizing baseline avidities could be even higher. This observation supports computer simulations suggesting that preexisting antibodies that mask immunodominant epitopes, such as the RBC-binding HA head domain, lead to the production of antibodies against less accessible epitopes such as the HA stalk domain [42]. This might be important for the generation of broadly neutralizing antibodies targeting the HA stalk domain, which show high potential *in vivo* despite poor neutralization activity *in vitro* [43, 44]. Further studies might investigate whether preexisting antibodies with high avidities against the HA head domain favor the production of HA-stalk antibodies.

Overall, we argue that our biophysical model of the HI assay not only generates detailed insights and hypotheses on influenza vaccine responses in small patient cohorts as here. Because the model requires only easy-to-establish measurements as inputs, we anticipate that it can also refine the analysis in larger vaccine studies, for instance, in screens to identify candidates for affinity maturation for subsequent detailed (but expensive) characterization by methods such as SPR.

## Methods

### Ethics statement

The study was conducted in accordance with the Declaration of Helsinki and approved by the Ethic committee northwest and central Switzerland (EKNZ ID 2014–141). All patients signed informed consent.

### Patient sera

Adult patients that received allogeneic hematopoietic stem cell transplantation (HSCT) at least one year before were recruited in a prospective cohort study in Switzerland (at the University Hospital Basel and the Cantonal Hospitals in Aarau and Lucerne) between October 2014 and January 2015 [45]. Only patients without known vaccine intolerance such as egg protein allergy or vaccine-associated adverse events were eligible for participation. In total, 57 patients were recruited; we included 45 of them in the present study based on the availability of serum samples. Following the standard of care for HSCT patients, each patient received two doses of

non-adjuvanted trivalent influenza vaccine (TIV), where the second dose was given 30 days after the first. Serum samples were collected prior to first vaccination (d0) and afterwards (d7, d30, d60, d180) and stored in aliquots at -80˚C. Almost all patients were in complete remission (42/45, 93%), and no patient showed progression. Patient characteristics are summarized in Table 2.

## Vaccine composition

Patients received two doses of the non-adjuvanted 2014/2015 trivalent influenza vaccine (Agrippal, Novartis, Switzerland), comprising inactivated, subunit influenza virus with 15 μg HA antigen of each vaccine strain: A/California/7/2009 (H1N1pdm09), A/Texas/50/2012 (H3N2) and B/Massachusetts/2/2012 (Yamagata lineage).

## HI assay

HI assays were performed according to the WHO manual [4]. Sera were pretreated with receptor destroying enzyme (RDE) (Sigma-Aldrich, C8772) and two-fold serially diluted, covering dilutions from 1:8 to 1:2048. A 0.75% (v/v) suspension of chicken RBCs (Cedarlane, CLC8800) and 4 HA units of influenza H1N1pdm09 virus (NYMC-X181) were used to perform the assay. The reported HI titer is the dilution factor of the highest serum dilution that showed full hemagglutination inhibition. The protocol has been published in detail [9].

## ELISA for influenza-specific IgG detection

ELISA 96-well plates (Thermo Scientific, 442404) were coated with 0.5 μg/mL whole virus H1N1pdm09 (NYMC-X181, 45 μg HA/mL) at 4˚C overnight. Plates were blocked with 5% bovine serum albumin (BSA) in PBS for 1h at room temperature (RT). Patient serum samples were 1:4000 diluted in 0.5% BSA in PBS. Reference serum was 1:1000 diluted (top dilution of calibration curve) and then six times four-fold serially diluted, yielding a calibration curve with seven measurements. After blocking and washing with 0.05% TWEEN 20 in PBS, 100 μL/well of diluted serum samples were added and incubated for 2h at RT. Unbound serum antibodies were removed by washing the plates four times, and bound serum IgG was detected by 70 μL/well of 1:3000 diluted rabbit anti-human IgG antibody linked to horseradish peroxidase (Agilent, P021402–2) incubated for 2h at RT. After washing, plates were developed with 100 μL/well TMB substrate solution (BD, 555214) for 15 min and stopped with 50 μL/well 2N $H_2SO_4$. Absorbance was measured at 450 and 620 nm. Measurements were background- and blank-corrected. To obtain a calibration curve, reference measurements were fitted using a four-parameter logistic equation (log concentration vs log absorbance). All measurements were performed in duplicates.

## Urea elution assay to measure IgG avidities

The ELISA described above was adapted to measure serum IgG avidities against influenza H1N1pdm09. Each serum was accordingly diluted to obtain a final concentration within the linear range of the calibration curve. After incubation with serum and washing as described above, each well was incubated for an additional 3h at RT with either 100 μL/well 4M urea (treated) or 100 μL/well PBS (untreated). The concentration of bound IgG was determined using a calibration curve, as described above. The fraction of IgG remaining bound after urea treatment compared to the untreated wells is reported as the avidity index. Avidities were determined in two experiments, each performed in duplicates.

## Model derivation

**Assay step 1: Binding of antibodies to virus.** We model the formation of antibody-epitope complexes, denoted by $C$, as a diffusion-controlled reaction between viruses and antibodies, following the model of antibody-virus interaction proposed by Groth (1963) [14]. For complex formation, free antibodies, $A$, need to successfully collide with free influenza virus particles, $V$. In addition, antibody-epitope complex formation depends on the probability of epitopes being unbound, denoted by $\phi$. The dynamics of complex formation is thus given by:

$$\frac{dC(t)}{dt} = k_{\text{ass}} \cdot A(t) \cdot V(t) \cdot \phi(t) - k_{\text{diss}} \cdot C(t) \ ,$$

where $k_{\text{ass}}$ and $k_{\text{diss}}$ are kinetic rate constants for association and dissociation, respectively. Note that some IgG antibodies bind bivalently to hemagglutinin, resulting in higher antibody affinities compared to their monovalent Fab fragments due to lower macroscopic dissociation rates [46, 47]. This antibody valency is lumped into the macroscopic dissociation constant $k_{\text{diss}}$.

The total number of epitopes is proportional to the total virus concentration $V_{\text{tot}_1}$ (where '1' indicates the first step of the assay), the average number of hemagglutinin receptors per virus, $r$, and the number of identical binding sites per hemagglutinin, $e$ ($e = 3$ since hemagglutinin is a homotrimer). With $e^*$ being the number of epitopes bound or shaded by one antibody molecule, the fraction of unbound epitopes is:

$$\phi(t) = \frac{e \cdot r \cdot V_{\text{tot}_1} - e^* \cdot C(t)}{e \cdot r \cdot V_{\text{tot}_1}} \ .$$

We assume that cross-linking of virus particles by antibodies is rare for the considered concentrations, such that the concentration of virus particles $V$ remains approximately the same during the experiment, i.e., $V \approx V_{\text{tot}_1}$. In addition, the mass balance for antibodies is $A_{\text{tot}_1} = A(t) + C(t)$. Substituting into the dynamics of complex formation leads to:

$$\frac{dC(t)}{dt} = \frac{k_{\text{ass}}}{e \cdot r} \cdot [A_{\text{tot}_1} - C(t)] \cdot [e \cdot r \cdot V_{\text{tot}_1} - e^* \cdot C(t)] - k_{\text{diss}} \cdot C(t) \ .$$

Since the average number of epitopes per virus particle $e \cdot r$ is constant, the dynamics is equivalent to a reversible bimolecular reaction following mass action kinetics with apparent dissociation constant $K_D^{\text{app}} = e \cdot r \cdot \frac{k_{\text{diss}}}{k_{\text{ass}}}$. We assume that antibody-virus binding is fast, such that after the incubation time the system is at steady-state. At steady-state, the complex concentration $C^{\text{eq}}$ fulfills

$$0 = [A_{\text{tot}_1} - C^{\text{eq}}] \cdot [e \cdot r \cdot V_{\text{tot}_1} - e^* \cdot C^{\text{eq}}] - K_D^{\text{app}} \cdot C^{\text{eq}} \ .$$

We exploit the analytic solution to this quadratic equation in $C^{\text{eq}}$ to compute the fraction of covered hemagglutinin epitopes at equilibrium, $\theta$, defined as:

$$\theta = \frac{e^* \cdot C^{\text{eq}}}{e \cdot r \cdot V_{\text{tot}_1}}$$

to obtain:

$$\theta = \frac{erV_{\text{tot}_1} + e^*A_{\text{tot}_1} + K_{\text{D}}^{\text{app}}}{2erV_{\text{tot}_1}} -$$

$$\frac{\sqrt{e^2r^2V_{\text{tot}_1}^2 + (K_{\text{D}}^{\text{app}})^2 + e^{*2}A_{\text{tot}_1}^2 - 2erV_{\text{tot}_1}e^*A_{\text{tot}_1} + 2K_{\text{D}}^{\text{app}}erV_{\text{tot}_1} + 2K_{\text{D}}^{\text{app}}e^*A_{\text{tot}_1}}}{2erV_{\text{tot}_1}} \quad . \tag{1}$$

**Assay step 2: Hemagglutination.** When RBC suspension is added to the system, two processes happen simultaneously: viruses bind to SA-linked receptors on RBCs with their free hemagglutinin binding sites, and RBCs stick together and form aggregates whenever they collide such that virus particles are able to cross-link them.

For virus binding to SA-linked receptors, we assume mass-action kinetics, leading to:

$$\frac{dV(t)}{dt} = -k_{\text{ass}}^{\text{RBC}} \cdot \underbrace{(1-\theta) \cdot r \cdot V(t)}_{\text{free virus sites}} \cdot \underbrace{[1-\rho(t)] \cdot b \cdot RBC_{\text{tot}_2}}_{\text{free RBC sites}} + k_{\text{diss}}^{\text{RBC}} \cdot \underbrace{\frac{b}{b^*} \cdot \rho(t) \cdot RBC_{\text{tot}_2}}_{\text{bound RBC sites}} \quad . \tag{2}$$

Kinetic constants for association and dissociation are denoted as $k_{\text{ass}}^{\text{RBC}}$ and $k_{\text{diss}}^{\text{RBC}}$. We assume $e = e^* = 3$ [17] to define the contribution of the concentration of hemagglutinin receptors that are not covered by antibodies (free virus sites) to the association rate. Association further depends on the amount of RBC binding sites that are not yet covered by virus, defined by the fraction of covered sites, $\rho(t)$, the average number of SA-linked surface receptors each RBC carries, $b$, and the total concentration of RBCs in step 2 of the assay, $RBC_{\text{tot}_2}$. For the dissociation term, we assume that one virus particle covers on average $b^*$ binding sites, since influenza virus particles are approximately 60-times smaller than RBCs (see below). The correction by $b^*$ reflects the definition of the fraction of covered RBC binding sites (making the term for bound RBS sites equivalent to the concentration of bound virus, $V_{\text{tot}_2} - V(t)$):

$$\rho(t) = \frac{b^* \cdot [V_{\text{tot}_2} - V(t)]}{b \cdot RBC_{\text{tot}_2}} \quad . \tag{3}$$

To capture RBC aggregation, let $B_k$ denote the concentration of agglutinating particles (individual RBCs and RBC aggregates) consisting of $k$ cells, with a maximum aggregate size $N$. To describe the dynamics, we use the Smoluchowski coagulation equation [24], where the rate of agglutination is proportional to an agglutination rate constant $k_{\text{agg}}$ and the number of available cross-linking sites $\rho(t)(1-\rho(t))(1-\theta)^2$, which is proportional to the number of mutual pairs of free binding sites on colliding RBCs and can be interpreted as a cross-linking probability:

$$\frac{dB_k(t)}{dt} = k_{\text{agg}}\rho(t)(1-\rho(t))(1-\theta)^2\left(\frac{1}{2}\sum_{i+j=k}K_{ij}B_i(t)B_j(t) - B_k(t)\sum_{i=1}^{N}K_{ik}B_i(t)\right) \quad .$$

For the special case $K_{ij} = K_{ik} = K$, where the kernel is independent of the particle size, there is a simple analytical solution for the discrete size distribution of aggregates. Let $\sum_{i=1}^{N}B_i(t) = B_N(t)$ denote the total concentration of particles, and $B_N(t=0) = RBC_{\text{tot}_2}$ the concentration of particles before agglutination. In addition, from mass conservation follows:

$\sum_{k=1}^{N} k B_k(t) = RBC_{\text{tot}_2}$. Summing over all values of $k$ then yields:

$$
\begin{aligned}
\frac{dB_N(t)}{dt} &= k_{\text{agg}}\rho(t)(1 - \rho(t))(1 - \theta)^2 \left[ \left(\frac{K}{2}\right) \cdot RBC_{\text{tot}_2}^2 - K \cdot RBC_{\text{tot}_2}^2 \right] \\
&= -k_{\text{agg}}\rho(t)(1 - \rho(t))(1 - \theta)^2 \left(\frac{K}{2}\right) \cdot RBC_{\text{tot}_2}^2 \ .
\end{aligned}
$$

Integrating once gives:

$$
B_N(t) = \frac{RBC_{\text{tot}_2}}{1 + \left(\dfrac{k_{\text{agg}}}{2}\right) \rho(t)(1 - \rho(t))(1 - \theta)^2 RBC_{\text{tot}_2}\ t} \ . \tag{4}
$$

Here, we set $K = 1$ such that the effect of $K$ is lumped into $k_{\text{agg}}$ because we estimated $k_{\text{agg}}$ from data (see below).

**Assay step 3: Determination of HI titer.** We define the degree of hemagglutination as:

$$
h(t) = \left( 1 - \frac{B_N(t)}{RBC_{\text{tot}_2}} \right) \cdot 100 \ , \tag{5}
$$

such that it takes values between 0% and 100%. If there is no hemagglutination, the concentration of agglutinated particles is the same as the initial concentration of RBCs ($B_N(t) = RBC_{\text{tot}_2}$) and the degree of hemagglutination is 0%. If all RBCs are agglutinated, there is only one agglutinating particle in the system and $B_N(t) = 1/N_A \cdot 10^9$ nM, where $N_A$ is Avogadro's number. Since $N_A \approx 6 \cdot 10^{23}$, $B_N \approx 10^{-15} \approx 0$ nM such that $h(t) = 100\%$.

## Model implementation

To obtain the degree of hemagglutination $h(t)$ in Eq 5, we compute $\theta$ from Eq 1, $\rho(t)$ for any time point $t$ in assay step 2 from Eqs 2 and 3, and the corresponding $B_N(t)$ from Eq 4.

In addition, the total concentration of antibodies is given by

$$
A_{\text{tot}_1} = 0.5 \cdot d_p \cdot d^j \cdot A_0 \ ,
$$

where $A_0$ is the initial serum antibody concentration, $d_p$ is the serum predilution factor, $d$ the serial dilution factor and $j$ the considered dilution step. The total concentrations of virus are

$$
V_{\text{tot}_1} = 0.5 \cdot V_0 \ \text{and} \ V_{\text{tot}_2} = 0.5 \cdot V_{\text{tot}_1},
$$

because each assay step involves adding equal volumes of solution; $V_0$ is the initial virus concentration. Analogously,

$$
RBC_{\text{tot}_2} = 0.5 \cdot RBC_0 \ ,
$$

where $RBC_0$ is the initial concentration of RBCs.

The model is implemented in the R package `himodel` (https://gitlab.com/csb.ethz/himodel).

## Model parameters and initial conditions

All model parameters and initial conditions could be either extracted or derived from literature (summarized in Table 1), except for the agglutination rate of RBCs ($k_{\text{agg}}$), which we estimated from data as described below.

**RBC concentration ($RBC_0$).** Following the WHO protocol [4], a 0.75% (v/v) suspension of chicken RBCs is used to measure HI titers against H1N1pdm09. This corresponds approximately to $1.875 \cdot 10^6$ cells/mL [48]. Given that 1 mol corresponds to $6.022 \cdot 10^{23}$ cells, the molar concentration is approximately $RBC_0 = 3.1 \cdot 10^{-5}$ nM. To determine the effect of pipetting errors, we set the RBC concentration range in the sensitivity analysis to 0.375%–1.5% (v/v) suspension, which corresponds to approximately $1.6 \cdot 10^{-5}$–$6.3 \cdot 10^{-5}$ nM.

**Number of sialic acid-linked receptors on RBC ($b$).** Influenza hemagglutinin binds to SA-linked surface receptors of RBCs. Human H1 influenza viruses bind preferentially to $\alpha2 \rightarrow 6$ linked SA [49], which occurs on the surface of chicken RBCs mainly in N-linked glycans [30, 31]. Chicken RBCs contain a mixture of $\alpha2 \rightarrow 3$ and $\alpha2 \rightarrow 6$-linked glycans in a ratio of approximately 60:40–50:50 [30]. The total number of N-glycan on RBCs has been estimated to be $1 \cdot 10^6$ [31]. Thus, we assume that the average number of receptors that can interact with hemagglutinin is $0.45 \cdot 10^6$. Chicken RBCs also have SA-linked O-glycans such as glycophorins [50] on their surface, but most of them contain $\alpha2 \rightarrow 3$-linked SA. Therefore, we neglect them.

**Steric virus factor ($b^*$): Number of sialic acid-linked receptors covered by bound virions.** Influenza virions are approximately 60-times smaller than RBCs [51]. To model the binding of virions to RBCs, we need to take into account that bound virions cover multiple SA-linked receptors. We estimated the average number of covered $\alpha2 \rightarrow 6$ SA-linked receptors, $b^*$, from simple geometry. We assume that SA-linked receptors are uniformly distributed on RBCs. Their estimated surface area ranges from 140–160 $\mu m^2$ and we assume an average surface area of $A_{RBC} = 150 \cdot 10^6$ $nm^2$ [52, 53]. The virus-covered area is determined by the virus' diameter. Most influenza virions are spherical with a diameter ranging from 84–170 nm and mean diameter $d = 120$ nm [51]. We estimate the shaded area from the circle area, which yields:

$$b^* = \frac{\pi(d/2)^2}{A_{RBC}} \cdot b \approx 34 \ . \tag{6}$$

In the sensitivity analysis, we sample $b^*$ assuming $d \sim \text{Unif}(84, 170)$, $A_{RBC} \sim \text{Unif}(130 \cdot 10^6, 170 \cdot 10^6)$, and $b \sim \text{Unif}(0.4 \cdot 10^6, 0.5 \cdot 10^6)$, where $d$ has unit nm, $A_{RBC}$ has unit $nm^2$ and $b$ is unitless.

**Virus concentrations ($V_0$).** To ensure the reproducibility of the HI assay, the same amount of virus particles must be used in each experiment. Therefore, virus concentration is measured in HA units, an operational unit that is determined in the so called HA titration assay, where virus is titrated against a constant amount of RBCs (same amount as used in the HI assay, i.e. 50 µL of 0.75% (v/v) RBC suspension are added to 50 µL serum-virus dilution). The amount of virus that agglutinates an equal volume of standardized RBC suspension is defined as 1 HA unit [4]. Electron microscopy data show that partial hemagglutination occurs at 1:1 binding (on average, one virus particle binds to one RBC) [48]. We assume that full hemagglutination requires at least 2:1 binding. We used the rate equation for virus-RBC binding (Eq 2) to determine the virus concentration that leads to 2:1 binding with $0.5 \cdot 3.1 \cdot 10^{-5}$ nM RBC (S1A Fig): $3.2 \cdot 10^{-5}$ nM. Assuming that this virus concentration corresponds to 1 HA unit in our model simulations, 4 HA units are approximately $V_0 = 1.3 \cdot 10^{-3}$ nM. In the sensitivity analysis, we varied $V_0$ in the range of 3–7 HA units.

**Agglutination rate ($k_{agg}$).** We inferred the agglutination rate of RBCs from hemagglutination inhibition experiments with a serum sample from a healthy volunteer that also served as our reference serum for the ELISA experiments (see below). We applied the inference procedure described in the next section. We used a broad uniform prior for $k_{agg} \sim \text{Uniform}(10^5, 10^9)$, set the coagulation kernel $K$ to 1 and fixed all remaining parameters to the values in

Table 1. The $k_{\mathrm{agg}}$ posterior distribution was approximately log-normal (centered at around $2 \cdot 10^6\ s^{-1}$ with 95% credibility interval of approximately $0.4 \cdot 10^6$–$13 \cdot 10^6\ s^{-1}$) with slightly heavier tail towards larger $k_{\mathrm{agg}}$ values since hemagglutination reaches saturation at approximately 30 min (S1B Fig). Data at earlier time points would be needed to infer $k_{\mathrm{agg}}$ with higher precision. We set $k_{\mathrm{agg}} = 2 \cdot 10^6\ s^{-1}$; the precision suffices as we are interested in hemagglutination at $\geq 30$ min.

## Inference of neutralizing antibody avidities

Given a measured IgG antibody concentration $A_i$ of serum sample $i$ (with estimated log mean $\mu_{A,i}$ and log standard deviation $\sigma_{A,i}$) and the corresponding HI titer determined in an HI assay with $j = 1, \ldots, J$ dilution steps, serum predilution factor $d_p$, and serial dilution factor $d$, the generative model to infer the posterior distributions for $K_{D,i}^{\mathrm{app}}$ is defined as follows:

$$
\begin{aligned}
K_{D,i}^{\mathrm{app}} &\sim \mathrm{Lognormal}(\mu_K,\ \sigma_K^2) \\
A_i &\sim \mathrm{Lognormal}(\mu_{A,i},\ \sigma_{A,i}^2) \\
A_{0,ij} &= A_i \cdot d_p \cdot d^j \\
\theta_{ij} &= f_\theta(A_{0,ij},\ K_{D,i}^{\mathrm{app}}) \\
\rho_{ij} &= f_\rho(\theta_{ij}) \\
h_{ij} &= f_h(\rho_{ij},\ \theta_{ij}) \\
p_{ij} &= \mathrm{logit}^{-1}(\alpha(h_{ij} - h_0)) \\
y_{ij} &\sim \mathrm{Bernoulli}(p_{ij}).
\end{aligned}
$$

Here, $A_{0,ij}$ is the final concentration of diluted serum IgG at dilution step $j$. It gives rise to sample- and dilution-specific $\theta_{ij}$, $\rho_{ij}$, and $h_{ij}$ as defined by Eqs 1, 3 and 5 (here abbreviated for convenience with $f_\theta$, $f_\rho$ and $f_h$ and with time dependencies dropped).

To determine the HI titer, each serum dilution $j$ is inspected for hemagglutination inhibition, and the reciprocal value of the minimal dilution that shows full inhibition is the HI titer. We treat the binary decision at each dilution step (inhibition/no inhibition) as a Bernoulli process with inhibition probability $p_{ij}$, a shorthand notation for $P(y_{ij} = 1 \mid h_{ij})$. The indicator variable $y_{ij}$ takes the value 0 if the hemagglutination degree $h_{ij}$ is above a certain threshold $h_0$ (no inhibition) and 1 otherwise (inhibition):

$$
y_{ij} = \begin{cases} 0, & \text{if } h_{ij} > h_0 \ (\text{no inhibition})\ , \\ 1, & \text{if } h_{ij} \leq h_0 \ (\text{inhibition})\ . \end{cases}
\tag{7}
$$

This binary decision is modelled by a logistic function with steepness parameter $\alpha$ and inflection point $h_0$. The conditional likelihood for $\mathbf{y_i}^T = (y_{i1}, y_{i2}, \ldots, y_{iJ})$ over all $J$ dilutions is then given by a product of Bernoulli likelihoods:

$$
P(\mathbf{y_i} \mid K_{D,i}^{\mathrm{app}}, A_i) = \prod_{j=1}^{J} p_{ij}^{y_{ij}} \cdot (1 - p_{ij})^{(1 - y_{ij})}
\tag{8}
$$

and the full posterior is:

$$
P(K_{D,i}^{\mathrm{app}}, A_i \mid \mathbf{y_i}) = \frac{P(K_{D,i}^{\mathrm{app}})P(A_i)P(\mathbf{y_i} \mid K_{D,i}^{\mathrm{app}}, A_i)}{P(\mathbf{y_i})}\ .
\tag{9}
$$

We sampled posterior distributions using the Metropolis-Hastings algorithm [54] with 6000 iterations, burn-in size of 1000 samples, and 5 chains. We used a broad log-normal prior for $K_{D,i}^{app}$ centered at 1 nM with log mean $\mu_k = 0$ and log standard deviation $\sigma_K = 4$. To define the value of $h_0$, we investigated the relationship between HA units and hemagglutination degree in our HA titration simulations. The model predicted that the hemagglutination degree is >75% for ≥1 HA unit (S1C Fig), which by definition corresponds to full hemagglutination. Thus, assuming symmetry, we consider $h_0 = 25\%$ a reasonable estimate, also assuming that differences below 25% cannot be distinguished by eye. However, a different value for $h_0$ does not affect the interpretation of our results: it would shift the posterior distribution of all samples either towards lower avidities (for larger $h_0$) or higher avidities (for smaller $h_0$). The steepness parameter $\alpha$ affects the width of the posterior distribution. Here, we set $\alpha = 15$ and then investigated the relationship between posterior distribution and resulting HI titer by sampling. We sampled 500 times from the joint posterior distribution of $K_{D,i}^{app}$ and $A_i$ for all patient sera $i$ and predicted the resulting HI titer to investigate the uncertainty in $K_{D,i}^{app}$ due to discretization of HI titer measurements and ELISA measurement error. On average, approximately 55% of samples resulted in the observed HI titer, whereas approximately 95% of samples also included HI titers one dilution step higher or lower than the actually observed HI titer.

## Reference serum

The concentration of H1N1pdm09-specific IgG antibodies was determined in ELISA experiments relative to a reference serum collected from a healthy volunteer on day 7 after vaccination with 2014/2015 TIV (Agrippal, Novartis, Switzerland), showing an HI titer of 512. Since the absolute reference concentration could only be determined by mass spectrometry, which was not feasible in this study, we estimated the concentration based on reported H1N1pdm09-specific IgG concentrations in vaccinated healthy adults with similar HI titers [25]. We set the reference concentration to 100 μg/mL (670 nM), yielding an estimated avidity for the reference serum of 0.4–0.8 nM, consistent with observed affinities for post-vaccination serum IgG for H1N1pdm09 in healthy adults [22].

## Identification of patients with increase in avidity and increase in non-neutralizing IgG

For each inferred $K_D^{app}$ value, we identified the uncertainty interval due to ELISA measurement error and dichotomization in HI titers by sampling from the joint posterior distribution (see above) and considered non-overlapping intervals as a significant change in $K_D^{app}$. To detect patients that produced non-neutralizing IgG after vaccination, we identified patients that showed no increase in HI titer while showing an increase in serum IgG that resulted in a significant decrease in avidity (S4A Fig).

## Sensitivity analysis

Sobol sensitivity analysis attributes variance in model output to individual model input factors using variance decomposition [32]. Given $k$ model inputs, the total variance $V(y)$ in model output can be decomposed as:

$$V(y) = \sum_i V_i + \sum_i \sum_{j>i} V_{ij} + \ldots + V_{12\ldots k} \, ,$$

(10)

where $V_i = V(E(Y|x_i))$ is the variance with respect to the distribution of input factor $x_i$. The second-order interaction term $V_{ij} = V(E(Y|x_i, x_j)) - V_i - V_j$ captures the part of the effect of $x_i$ and

$x_j$ that is not described by the first order terms $V_i$, $V_j$ and so on. The relative contribution of each term to the unconditional variance $V(y)$ serves as a measure of sensitivity. For instance, $V_i$ will be large, if $x_i$ is influential. The first order Sobol sensitivity index is defined as

$$S_i = \frac{V_i}{V(y)} \quad . \tag{11}$$

To obtain the total contribution of $x_i$, that is the sum of all terms in the variance decomposition that include $x_i$, we compute the total contribution to variance $V(y)$ due to all factors but $x_i$, denoted by $\mathbf{x_{-1}}$. The total Sobol sensitivity index for $x_i$ is then given by

$$S_i^T = \frac{V(y) - V(E(y|\mathbf{x_{-1}}))}{V(y)} \quad . \tag{12}$$

We used Monte Carlo estimation to estimate Sobol indices [55, 56] implemented in the R package `sensitivity` [57] with n = 10000 random samples of model input vector $\mathbf{x^T} = (x_1, x_2, \ldots, x_k)$ and 10 bootstrap replicates to estimate confidence intervals. Input variables were assumed to be independent of each other. We considered $k = 12$ inputs sampled within a biologically reasonable range (Table 1).

## Statistical analysis

Serum IgG and inferred $K_D^{\mathrm{app}}$ values were available for 43 patients at five time points ($t = 0, 7, 30, 60, 180$ days) with 197 observations in total. To estimate the effects of a patient's immune state on serum IgG and avidity ($1/K_D^{\mathrm{app}}$), we used a linear mixed model with patient-specific random intercepts that takes the following general form:

$$
\begin{aligned}
y_{ij} &= \beta_0 + x_{ij}^T \beta_1 + \gamma_i + \epsilon_{ij}, \\
\gamma_i &\sim \mathcal{N}(0, \ \sigma_\gamma^2), \\
\epsilon_{ij} &\sim \mathcal{N}(0, \ \sigma_\epsilon^2),
\end{aligned}
$$

where $y_{ij}$ is the log2-transformed IgG concentration or $1/K_D^{\mathrm{app}}$ value, respectively, of patient $i$ at time point $j$, $x_{ij}$ is a $p$-dimensional vector of $p$ covariates, $\beta_0$ is an intercept term, $\beta_1$ is a vector of fixed effects, $\gamma_i$ the random patient-specific intercept, and $\epsilon_{ij}$ models the within-patient measurement error. We modeled the observed rise and fall of serum IgG and $1/K_D^{\mathrm{app}}$ value after vaccination using a second-degree polynomial. To distinguish time trends in avidity between neutralizing and non-neutralizing IgG responders, we added a dummy variable for neutralizing response when analyzing response in avidity. Time post HSCT $\leq$ 2 years, cGVHD grade, and immunosuppression grade were added as fixed effects on intercept to investigate effects on baseline, and on slope to investigate effects on response. Time post HSCT $\leq$ 2 years was encoded as a binary variable (1 for $\leq$ 2 years, 0 for > 2 years). Both cGVHD and immunosuppression grade were encoded as numerical variables with values 0, 1, 2, 3, such that grade 0 is the reference level, and there is a linear increase in effect with increasing grade. To control for potential confounders, we corrected for sex and age. For model selection, the full model with fixed effects on slope and intercept was fitted using maximum likelihood estimation implemented in the `lmer4` package [58] and type II ANOVA by Satterthwaite's approximation provided by the `lmerTest` package [59].

We detected no significant effects on response and therefore removed the fixed effects on slope and refitted the final models using restricted maximum likelihood estimation to obtain unbiased estimates [58]. Residuals indicated that the normality assumption was satisfied (S4B Fig). Confidence intervals were computed via the Wald method provided by `lme4`. To

compare the results with HI titers, we estimated the effect of time post HSCT $\leq$ 2 years, cGVHD grade, and immunosuppression on HI titers controlling for age, sex, and time after vaccination using a generalized linear regression model for sequential, ordered data [34]. The model was fitted using maximum likelihood estimation implemented in VGAM [60].

## Supporting information

**S1 Fig. Simulation results for the HA titration assay with influenza H1N1pdm09 and model sensitivity.** (**A**) Binding kinetics of virus particles to red blood cells. We assume that full hemagglutination requires at least two bound virus particles per cell. (**B**) Hemagglutination kinetics. (**C**) For HA units $\geq 1$, the hemagglutination degree is $> 75\%$, which is by definition interpreted as full hemagglutination. Gray areas and error bars indicate the uncertainty due to uncertainty in model parameters. (**D**) Performing the HI assay with 4 HA units balances sensitivity and robustness. There is a clear distinction between inhibition and no inhibition. (**E**) In addition, the assay detects with 4 HA units lower antibody concentrations than with $\geq 8$ HA units.
(TIF)

**S2 Fig. Marginal posterior distributions of the apparent dissociation constants in 43 patients (197 posteriors in total).** Some posteriors show larger variance due to larger measurement error in ELISA-detected IgG concentration. Here, for samples with HI titer $< 8$, the shown posterior distributions correspond to the inferred avidity when assuming HI titer = 4 (affected 23 serum samples from seven patients).
(TIF)

**S3 Fig. HI titer, ELISA-detected anti-H1N1pmd09 serum IgG concentration, inferred apparent dissociation constant and experimentally determined avidity index in twelve patients.** Avidity indices correspond to the fraction of H1N1pmd09-specific serum IgG remaining bound after 4M urea treatment. Data show mean and standard deviation for serum IgG and avidity indices and the median of the posterior distribution with the uncertainty range due to discretized HI titer measurements and ELISA measurement error for inferred apparent dissociation constants $K_D^{app}$. Most patients showed either little or no increase in avidity. In some patients, the measured avidity decreased and then returned back to baseline on d180, potentially because the vaccine-induced short-lived antibodies were more sensitive to urea treatment, resulting in antibody denaturation.
(TIF)

**S4 Fig. Vaccine response analysis in 43 patients (197 samples in total).** (**A**) Fold change in inferred avidity and serum IgG concentration after vaccination. Error bars indicate uncertainty in fold change due to uncertainty in inferred $K_D^{app}$-values. Shading indicates regions with qualitatively different responses to vaccination. (**B**) Residual plots of the regression models used to investigate associations of criteria for compromised immune response with avidity and serum IgG concentration.
(TIF)

**S5 Fig. Inferred avidity, serum IgG concentration, and HI titers by time after transplantation, immunosuppression grade, and cGVHD grade in 43 patients (197 samples in total).** Note that data show one-dimensional associations, whereas regression analysis was performed with a high-dimensional model simultaneously accounting for time after transplantation, immunosuppression/cGVHD grade, and correcting for sex and age.
(TIF)

## Acknowledgments

We thank our clinical collaborators Nathan Cantoni (Cantonal Hospital Aarau) and Sabine Ruosch-Girsberger (Cantonal Hospital Lucerne) for providing patient sera and supporting the cohort study. We thank Lukas Kaufmann and Dominik Vogt for their excellent technical support with patient samples, and Fabian Rudolf for advice on the experimental design of avidity measurements.

## Author Contributions

**Conceptualization:** Janina Linnik, Adrian Egli, Jörg Stelling.

**Data curation:** Janina Linnik, Yvonne Hollenstein.

**Formal analysis:** Janina Linnik.

**Funding acquisition:** Adrian Egli, Jörg Stelling.

**Investigation:** Janina Linnik, Mohammedyaseen Syedbasha, Yvonne Hollenstein, Jörg Halter.

**Methodology:** Janina Linnik, Jörg Stelling.

**Project administration:** Jörg Halter, Adrian Egli, Jörg Stelling.

**Resources:** Adrian Egli, Jörg Stelling.

**Software:** Janina Linnik.

**Supervision:** Adrian Egli, Jörg Stelling.

**Validation:** Janina Linnik.

**Visualization:** Janina Linnik.

**Writing – original draft:** Janina Linnik, Jörg Stelling.

**Writing – review & editing:** Mohammedyaseen Syedbasha, Yvonne Hollenstein, Jörg Halter, Adrian Egli.

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
