## [Decision Letter · Decision Letter 0]

11 Oct 2021

Dear Stelling,

Thank you very much for submitting your manuscript "Model-based inference of neutralizing antibody avidities against influenza virus" for consideration at PLOS Pathogens. As with all papers reviewed by the journal, your manuscript was reviewed by members of the editorial board and by several independent reviewers. The reviewers appreciated the attention to an important topic. Based on the reviews, we are likely to accept this manuscript for publication, providing that you modify the manuscript according to the review recommendations.

Sincerely,

Shin-Ru Shih

Section Editor

PLOS Pathogens

Shin-Ru Shih

Section Editor

PLOS Pathogens

Kasturi Haldar

Editor-in-Chief

PLOS Pathogens

orcid.org/0000-0001-5065-158X

Michael Malim

Editor-in-Chief

PLOS Pathogens

orcid.org/0000-0002-7699-2064

Reviewer Comments (if any, and for reference):

Reviewer's Responses to Questions

**Part I - Summary**

Reviewer #1: The hemagglutination inhibition assay is the gold standard for assessing the immune status to influenza. While there are other immunological assays that can be used, the ease and wide-spread use of the HI assay can be the basis for predictive, model-based analyses. Here, Linnik and colleagues demonstrate that the apparent IgG dissociation constant and serum IgG concentration are most predictive of HI readout. Moreover, avidity was inferred from measured HI titers and IgG concentration from samples performed from a prospective cohort. The manuscript is well-written and explains the HI assay quite well – even for non-virologists.

Reviewer #2: The manuscript by Linnik et al describes a biophysical model of the hemagglutination inhibition (HI) assay to determine both the concentration and avidity of neutralizing antibodies present in serum. Such a model would be useful for the field, since these parameters can typically only be determined using separate assays (e.g. ELISA). The authors train a model using experimental data and they test their model on patient samples. They find promising correlations in the second part of the manuscript using sera from vaccinated hematopoietic stem cell transplantation (HSCT) patients (n=45). They further confirm their results by measuring the avidity using an enzyme-linked immunosorbent assay. Finally, the authors use their findings to suggest that immunosuppressive treatments could lead to a lower baseline avidity. Overall, the study establishes the use of biophysical model for the estimation of HA antibody avidity. The study represents an elegant methodological advancement that has a lot of potential. However, the manuscript contains a large number of assumptions and statements that are very confusing, and it would be useful if the authors could address these.

Reviewer #3: Linnik et al. presented a well-organized and well-written manuscript. This study focuses on a biophysical model establishment for inferring the level of antibody avidity via a comprehensive analysis of HI assay, IgG concentration and other related factors in HSCT patients. In addition, a R program package “himodel” was lunched online that can be utilized by other research teams doing vaccine studies. Since influenza virus infection is associated with high mobility and mortality in immunocompromised patients, refining the examination method for evaluating vaccine-induced immunity is as crucial as vaccine development. This study unveiled some insight information about HI assay and established a novel model for inferring antibody avidity which may accelerate the innovation of vaccine study. Authors also pointed out the limitation of the current model, which could remind other research teams avoiding possible bias.

**Part II – Major Issues: Key Experiments Required for Acceptance**

Reviewer #1: (No Response)

Reviewer #2: Major comments:

1. To establish their model, the authors generated an HI ‘training’ dataset using a monoclonal HA antibody and the 2009 pandemic H1N1 strain, and various parameters from the literature, such as the sialic acid-HA binding constant of the A/X-31 H3N2 strain. The authors claim that they used these parameters to “make the model more specific for H1N1 pdm09” (line 94/95). However, the H3N2 virus (note that X-31 is a recombinant A/Aichi/1968 virus) is a rather different virus than the pandemic H1N1 virus. Moreover, HA proteins from different influenza virus strains can have widely different Kd’s for different sialic acids and bind sialic acid in different ways (e.g. https://www.pnas.org/content/109/52/21474). These very basic points makes one wonder if the authors should not have spent some effort on characterizing the biochemical and biophysical properties of their HA and antibody combination, before making sweeping claims about the general applicability and robustness of their model.

2. When the authors use the model to infer the antibody concentration and avidity from patient samples, they fix the model parameters (line 140-141). It is unclear why they chose this approach, given that their model was trained using a monoclonal antibody and parameters that may not be appropriate for all serum samples. No experimental data is shown to justify this approach.

Reviewer #3: Line 218-225: Authors stated that the patients with HSCT > 2 years may have developed durable H1N1pmd09-neutralizing antibody; whereas the early transplant patients (HSCT < 2 years) with lower IgG concentration and HI titer was resulted in previous vaccinations did not induce (detectable) affinity maturation. However, early transplant patients might also receive vaccines before receive transplantation. In HSCT < 2 years patients, the vaccine-induced immunity may also be compromised by the onset of immunosuppressive treatment. These descriptions could be moved to the discussion section.

**Part III – Minor Issues: Editorial and Data Presentation Modifications**

Reviewer #1: The manuscript is generally straightforward – however there are other limitations to this as contemporary H3N2 viruses are known to have agglutination problems and thus this approach is arguably effective only when assessing the response towards H1N1s. Perhaps this should be briefly discussed or pointed on in the discussion.

Reviewer #2: Minor comments:

3. The authors use the output of the model to assess the quality of the model in Fig. 2. However, the authors do not use experimental data to verify the output of the model at this point. Indeed, Fig. 2b and c only show predicted data, but no comparison to experimental data and statistical testing is used to demonstrate how reliable the model is. It is thus unclear why the authors call their model “robust” (line 111 and 134). The comparison with the experimental data follows in Fig. 3, so these claims seem premature at this point in the text.

4. line 64: The authors report that they use a protease treatment of the serum sample to minimize unspecific antibody binding. This protease treatment is confusing, because it is not explained in the material and methods section. Instead, the authors describe treating the serum with a sialidase, which is a glycoside hydrolase and not a protease at all. What is the method that they used?

5. Figure 4c is not referenced in the main text.

Reviewer #3: -As the authors described in Methods (line 315-316) and line 142-143, all patients were immunized with two doses of trivalent seasonal influenza vaccine. The manuscript only presents specific immune response against H1N1pmd09. Please provide some discussion about the inferences of antibody avidity against other virus strains.

-In line 241-244, the authors stated that results from the model explained why the HI assay is the gold standard in serological studies and the limitations of the current model. In line 249-257, the authors also mentioned some limitations of the model and provide some resolutions for these limitations. Accordingly, it still needs techniques (SPR or calorimetry measurements) to overcome these limitations. The authors should provide some discussion to emphasize the advantages and impacts of the biophysical model in future vaccine studies.

PLOS authors have the option to publish the peer review history of their article (what does this mean?). If published, this will include your full peer review and any attached files.

Reviewer #1: No

Reviewer #2: No

Reviewer #3: No

Figure Files:

Data Requirements:

Reproducibility:

References:

---

## [Editor Report · Decision Letter 1]

3 Jan 2022

Dear Stelling,

We are pleased to inform you that your manuscript 'Model-based inference of neutralizing antibody avidities against influenza virus' has been provisionally accepted for publication in PLOS Pathogens.

Best regards,

Shin-Ru Shih

Section Editor

PLOS Pathogens

Shin-Ru Shih

Section Editor

PLOS Pathogens

Kasturi Haldar

Editor-in-Chief

PLOS Pathogens

orcid.org/0000-0001-5065-158X

Michael Malim

Editor-in-Chief

PLOS Pathogens

orcid.org/0000-0002-7699-2064
---

## [Editor Report · Acceptance letter]

26 Jan 2022

Dear Stelling,

We are delighted to inform you that your manuscript, "Model-based inference of neutralizing antibody avidities against influenza virus," has been formally accepted for publication in PLOS Pathogens.

Best regards,

Kasturi Haldar

Editor-in-Chief

PLOS Pathogens

orcid.org/0000-0001-5065-158X

Michael Malim

Editor-in-Chief

PLOS Pathogens

orcid.org/0000-0002-7699-2064